# A Comparison of DNA Metabarcoding and Microscopy Methodologies for the Study of Aquatic Microbial Eukaryotes

Ioulia Santi [1], Panagiotis Kasapidis [1], Ioannis Karakassis [2] and Paraskevi Pitta [3,*]

1 Hellenic Centre for Marine Research (HCMR), Institute of Marine Biology, Biotechnology and Aquaculture (IMBBC), P.O. Box 2214, GR71003 Heraklion, Crete, Greece; isanti@hcmr.gr (I.S.); kasapidi@hcmr.gr (P.K.)

2 Department of Biology, Voutes Campus, University of Crete, 70013 Heraklion, Crete, Greece; karakassis@uoc.gr

3 Hellenic Centre for Marine Research (HCMR), Institute of Oceanography, P.O. Box 2214, GR71003 Heraklion, Crete, Greece

* Correspondence: vpitta@hcmr.gr; Tel.: +30-2810-337-829

**Abstract:** The procedures and methodologies employed to study microbial eukaryotic plankton have been thoroughly discussed. Two main schools exist—one insisting on classic microscopy methodologies and the other supporting modern high-throughput sequencing (DNA metabarcoding). However, few studies have attempted to combine both these approaches; most studies implement one method while ignoring the other. This work aims to contribute to this discussion and examine the advantages and disadvantages of each methodology by comparing marine plankton community results from microscopy and DNA metabarcoding. The results obtained by the two methodologies do not vary significantly for Bacillariophyta, although they do for Dinoflagellata and Ciliophora. The lower the taxonomic level, the higher the inconsistency between the two methodologies for all the studied groups. Considering the different characteristics of microscopy-based identification and DNA metabarcoding, this work underlines that each method should be chosen depending on the aims of the study. DNA metabarcoding provides a better estimate of the taxonomic richness of an ecosystem while microscopy provides more accurate quantitative results regarding abundance and biomass. In any case, the combined use of the two methods, if properly standardized, can provide much more reliable and accurate results for the study of marine microbial eukaryotes.

**Keywords:** microbial eukaryotes; DNA metabarcoding; microscopy; marine plankton; 18S rRNA

## 1. Introduction

The use of molecular ecology methodologies for the study of plankton is standard and considerable progress has occurred as a result of their use [1]. The study of prokaryotic plankton using molecular methods has provided information that would have been impossible to obtain using microscopy or other analytical methods. For instance, by applying metagenomic methods, bacterial community composition was linked to specific substrate organic molecules [2], and using 16S DNA metabarcoding, specific Alphaproteobacteria were found to successfully evade grazing by larger organisms [3].

Studying planktonic eukaryotes using molecular ecology techniques has triggered a debate on the advantages and disadvantages of modern methods compared with classical methods and on the combination of such methodologies. Some of the issues include the necessity for the combination of the methodologies and what additional value this could provide to the understanding of the ecosystem [1,4,5].

Microscopic observation allows identifying and enumerating microorganisms based on their morphology in a fixated sample [6]. The groups most commonly determined and enumerated under the microscope are diatoms, dinoflagellates, ciliates, coccolithophorids, and flagellated cells [6,7]. How these groups correspond to taxonomic groups according to

the National Center for Biotechnology Information (NCBI) taxonomy is summarized in Table 1.

**Table 1.** Correspondence of microbial eukaryote groups identified using microscopy to taxonomic groups according to NCBI taxonomy.

| Microscopy Groups | Taxonomical Groups |
|---|---|
| Diatoms | Bacillariophyta (Stramenopiles) |
| Dinoflagellates | Dinoflagellata (Alveolata) |
| Ciliates | Ciliophora (Alveolata) |
| Coccolithophorids | Prymnesiophyceae (Haptophytes, Haptista) |
| Flagellated cells [1] | Bigyra/Cercozoa/Chlorophyta/Cryptophyta/Haptophyta/ Hyphochytriomycota/Labyrinthulomycetes/Ochrophyta/ Oomycota/Rhodophyta |

[1] The category of flagellated cells refers to cells that have flagella and are not diatoms, dinoflagellates, ciliates, or coccolithophorids. These cells may taxonomically belong to any of the listed groups.

Many researchers have mastered the taxonomy of photoautotrophic plankton (usually termed phytoplankton), but their expertise only pertains to representative species of phytoplankton groups, particularly to diatoms and dinoflagellates. However, the term phytoplankton is now considered outdated as the true functional roles of plankton are more complex than the obsolete categorization to phyto- and zooplankton and, at the moment, only diatoms are considered solely photoautotrophic plankton [8]. Even though it is clear nowadays that most microbial eukaryotes show mixotrophic behavior [9] and play pivotal roles in the water-column ecosystem [10–12], groups such as Ciliophora, Cercozoa, and Haptophyta are usually omitted in microscopy studies, as they are hard to identify under light microscopy. Still, most protocols, guidelines, manuals, and standard operating procedures only describe procedures and methodologies for enumerating phytoplankton cells [13–16].

There are detailed studies that provide a comparison between microscopy and metabarcoding for communities of plankton Metazoa [17–19], sediment Metazoa [20], freshwater periphyton [21], freshwater plankton [22] and estuarine plankton [23], as well as studies focusing on one planktonic group (for example, Bacillariophyta [24]), and others omitting microscopic examination of the whole microplankton (20–200 μm) [23,25,26]. The present study attempts to critically approach the debate concerning the two methodologies and to contribute to the ongoing discussion by directly comparing whole community microbial eukaryote diversity results from DNA metabarcoding and microscopy examination with the aim of answering the following questions: What are the advantages and disadvantages of each methodology and how are these reflected in the results? Do either of the two methodologies produce higher quality results when used for microbial eukaryotic communities? Under what circumstances may a method be selected over the other when studying microbial plankton eukaryotes? Could the two methodologies be combined to provide a more complete picture of microbial eukaryotic communities?

## 2. Materials and Methods

For this work, we used data from the analysis of seawater samples from a mesocosm experiment [27], from the open sea [28], and from coastal areas (I. Santi unpublished). The mesocosm experiment was performed using coastal seawater from the Eastern Mediterranean, which was enriched with nutrients at different concentrations. Open sea samples were collected from 4 sites in the Libyan Sea. Coastal seawater samples were collected from different sites near the Greek shore; all coastal sites had a 20 m maximum depth. Table 2 summarizes the samples that were used in this study. In total, 27 samples were analyzed using inverted microscopy and 18S DNA metabarcoding.

**Table 2.** List of samples used in this study and sampling details. Coordinates are presented in decimal degrees. The column labelled "Experimental Conditions" only refers to the mesocosm experiment samples.

| Sample Name | Project | Coordinates | Seawater Depth (m) | Sampling Date (DD/MM/YYYY) | Experimental Conditions |
|---|---|---|---|---|---|
| mes-1 | | | | 03/10/2014 | No treatment |
| mes-2 | | | | 27/09/2014 | Initial conditions |
| mes-3 | Mesocosm Experiment | 35.335° N 25.281° E | 2 | 09/10/2014 | High nutrient addition |
| mes-4 | | | | 27/09/2014 | Initial conditions |
| mes-5 | | | | 03/10/2014 | No treatment |
| mes-6 | | | | 09/10/2014 | Low nutrient addition |
| mes-7 | | | | 27/09/2014 | Initial conditions |
| mes-8 | | | | 03/10/2014 | Low nutrient addition |
| mes-9 | | | | 03/10/2014 | No treatment |
| mes-10 | | | | 09/10/2014 | High nutrient addition |
| coa-1 | | 40.770° N 23.813° E | 20 | 5/7/2014 | |
| coa-2 | | 40.835° N 25.744° E | 20 | 4/7/2014 | |
| coa-3 | Coastal Sampling | 37.931° N 23.684° E | 20 | 24/7/2014 | |
| coa-4 | | 38.508° N 23.517° E | 10 | 22/7/2014 | |
| coa-5 | | 40.915° N 24.566° E | 20 | 5/7/2014 | |
| coa-6 | | 40.770° N 23.813° E | 2 | 5/7/2014 | |
| coa-7 | | 40.835° N 25.744° E | 2 | 4/7/2014 | |
| coa-8 | | 37.931° N 23.684° E | 2 | 24/7/2014 | |
| coa-9 | | 38.508° N 23.517° E | 2 | 22/7/2014 | |
| oce-1 | | 34.667° N 24.367° E | 5 | 15/4/2016 | |
| oce-2 | | 34.250° N 25.483° E | 50 | 14/4/2016 | |
| oce-3 | Open Sea Sampling | 34.433° N 26.383° E | 75 | 10/4/2016 | |
| oce-4 | | 35.033° N 23.467° E | 5 | 17/4/2016 | |
| oce-5 | | 34.250° N 25.483° E | 5 | 14/4/2016 | |
| oce-6 | | 34.667° N 24.367° E | 50 | 15/4/2016 | |
| oce-7 | | 34.250° N 25.483° E | 75 | 14/4/2016 | |
| oce-8 | | 34.433° N 26.383° E | 5 | 10/4/2016 | |

Seawater for microscopy analyses was fixed with Lugol's solution and analyzed using an inverted microscope after Utermöhl-type sedimentation. Cells from the mesocosm experiment and the coastal areas were identified to the genus level and open sea samples

to the phylum level. For the calculation of biomass, dimensions were measured and a geometric shape was assigned to each cell [29,30].

Seawater for DNA metabarcoding was collected through filtration using a peristaltic pump. Six liters of the mesocosm experiment seawater were filtered through 0.2 μm polyethersulfone membrane filters. For the open sea samples, 21 L of seawater were filtered sequentially through 20 μm pore size nylon mesh membranes and 0.8 μm pore size polycarbonate membrane filters. The results from the different fractions were combined in this work so that plankton >0.8 μm was included in the open sea samples. Coastal seawater samples were filtered through 0.2 μm polyethersulfone filter membranes. The filtered volume of seawater for the coastal samples was 12 L. For all cases, a prefiltration step at 200 μm was preceded to exclude all organisms larger than 200 μm in size, and filtrations were completed within 75 min. All membrane filters were stored at −80 °C until analyses were performed.

DNA was extracted from the filters using the PowerWater DNA isolation kit (Qiagen) following the manufacturer's instructions with the modification of double elution of the DNA. The 18S rRNA gene was amplified using the specific primers described in [31]. The two-step parallel multiplexing approach was used, and the specific primers of the first-step polymerase chain reaction (PCR) included a short sequence tail complementary to the second-step PCR primers [32]. The second-step PCR primers included short index sequences for post-sequencing sample identification and flow cells adaptors for Illumina MiSeq. The first-step PCR was performed in 20 μL reactions (PCR mix: 10 ng DNA extract, 0.5 μM of forward and reverse primers, 0.03 U/μL Taq polymerase, 0.3 mM dNTPs mixture, and KAPA HiFi Fidelity Buffer) at 29 amplification cycles. The second-step PCR was performed in 20 μL reactions (PCR mix: 10 ng clean first-step PCR product, 1 μM of forward and reverse primers, 0.02 U/μL Taq polymerase, 0.3 mM dNTPs mixture, 200 mM trechalose, and KAPA HiFi Fidelity Buffer) at 5 amplification cycles. PCR products were purified using magnetic beads (Agencourt AMPure XP, Beckman Coulter, Indianapolis, IN, USA). Samples were quantified by gel, and equimolar amounts of each sample were pooled for sequencing. The MiSeq Reagent Kit v2 was used for sequencing on the Illumina MiSeq platform at the Institute of Marine Biology Biotechnology and Aquaculture of the Hellenic Centre for Marine Research, Greece. Raw sequences used in this study were submitted to the ENA-GenBank under the study accession numbers PRJEB25787, PRJEB26382, and PRJEB26396.

Raw sequence reads were processed using the OBITools metabarcoding software suite [33]. Sequences were trimmed to a median Phred quality score higher than 40, paired reads were assembled with at least 50 nucleotides overlapping, and pairs of higher than 40 alignment quality score were selected. The length filter was applied so that only sequences of length between 80 and 200 bp were selected. Sequences were de-replicated with subsequent removal of sequences occurring only once in the dataset. PCR and sequencing errors as well as chimeras were removed using the obiclean algorithm of OBITools that kept sequences not having related sequence counts of more than 5% of their count. Clustering to amplicon sequence variants (ASVs) was performed using SWARM (v2), a stepwise deterministic aggregation algorithm [34]. The taxonomy was assigned with the ecotag algorithm [33] using NCBI (https://www.ncbi.nlm.nih.gov/taxonomy accessed on 18 August 2017) as a taxonomic reference. Reference sequences were acquired from the EMBL-EBI databank (https://www.ebi.ac.uk/ accessed on 1 September 2017) and, subsequently, the 18S amplicon was targeted with in silico ecoPCR [35] using the primers used for the actual PCR [31]. Datasets were refined by blank correction, removal of false positive results due to random index swap [36], minimal abundance filtering (removal of OTUs with less than 5 reads), and contaminant removal [36]. Samples with fewer than 10,000 final sequence reads were removed from the dataset. The number of final sequencing reads ranged between 10,226 and 138,589.

Instead of relative abundance, the biomass measurement was selected and used for the microscopy analyses, as it is assumed that a larger cell is likely to contain more copies

of the 18S rRNA gene [37,38], and thus biomass would be more comparable to the DNA metabarcoding data. The biomass measurements for each sample were transformed to relative biomass (percentage of the total biomass per taxonomic group) in order to be directly comparable to the DNA metabarcoding data (i.e., relative abundance of each taxonomic group, estimated as the percentage of sequencing reads assigned to a taxonomic group compared to the total reads per sample). To check for differences in the community composition produced by the two methods, we applied generalized linear modeling based on beta distribution. The percentage of relative abundance of the eukaryotic phyla was the response variable (*y*) and was tested against the explanatory variables (*x*), methodology (i.e., metabarcoding and microscopy), and interaction of the methodology with each of the examined phyla. All statistical analyses and data processing were performed using R v4.0.3, the community ecology package vegan v2.5-6 [39] and the beta regression package betareg v3.1-3 [40].

## 3. Results and Discussion

The overall beta distribution model was significant (Table 3); however, the difference in methodology alone was not able to explain the variance in the composition percentages. The inclusion of the interaction in the model was significant based on a chi-square test at the 1% significance level ($p < 0.001$). This means that the methodology significantly explained the variance in a different way for each phylum; this was also obvious when examining the relative abundance estimates from metabarcoding and the relative biomass from microscopy for each phylum (Figure 1, Dataset S1). To the best of our knowledge, only one study reported high correspondence in the composition of estuarine microbial eukaryotic plankton when comparing measurements between microscopy and DNA metabarcoding [23].

Between the two techniques (microscopy and DNA metabarcoding), the percentages of Bacillariophyta differed negligibly (Figure 1A) in contrast with other studies; these studies, however, examined only the <45 μm plankton community [26], whereas here, a community of a larger size range was examined (i.e., <200 μm). Nevertheless, Bacillariophyta showed a higher presence in most metabarcoding results. The percentage of Dinoflagellates differed significantly between the two techniques for all samples (Figure 1B), and the percentages produced by metabarcoding were significantly higher than the microscopy percentages (Table 3). This is similar to the findings of Piredda et al. (2017). It seems that the broadly used fixatives (Lugol's, formaldehyde, and glutaraldehyde) cannot preserve the morphology of unarmored dinoflagellate forms and may cause cell disruption, especially in the nanosized spectrum (2–20 μm) [16,25]. These findings for dinoflagellates show that even for the well-studied groups of eukaryotic microorganisms (Dinoflagellata), the morphological identification exclusively may lead to false classification [25].

**Table 3.** Generalized linear modeling based on beta distribution. The eukaryotic phyla percentages were modeled against the analysis method (metabarcoding and microscopy) and the interaction between the analysis method and the specific phylaanalyzed. Estimations were calculated based on maximum likelihood. Mbc: metabarcoding; Msp: microscopy; Other Flag: other flagellated cells.

|  | Estimate | Std. Error | z Value | *p*-Value |
|---|---|---|---|---|
| (intercept) | −1.062 | 0.188 | −5.655 | <0.001 |
| Method | −0.444 | 0.283 | −1.571 | 0.116 |
| Mbc: Ciliophora | −1.457 | 0.293 | −4.972 | <0.001 |
| Msp: Ciliophora | 0.645 | 0.290 | 2.227 | 0.026 |
| Mbc: Dinofl. | 1.022 | 0.255 | 4.005 | <0.001 |
| Msp: Dinofl. | 0.543 | 0.291 | 1.865 | 0.062 |
| Mbc: Other Flag. | −0.088 | 0.264 | −0.334 | 0.739 |
| Msp: Other Flag. | 0.170 | 0.306 | 0.556 | 0.578 |
| phi coefficient of beta distribution | 5.242 | 0.587 | 8.93 | <0.001 |
| Model R$^2$: 0.390 | | | | |

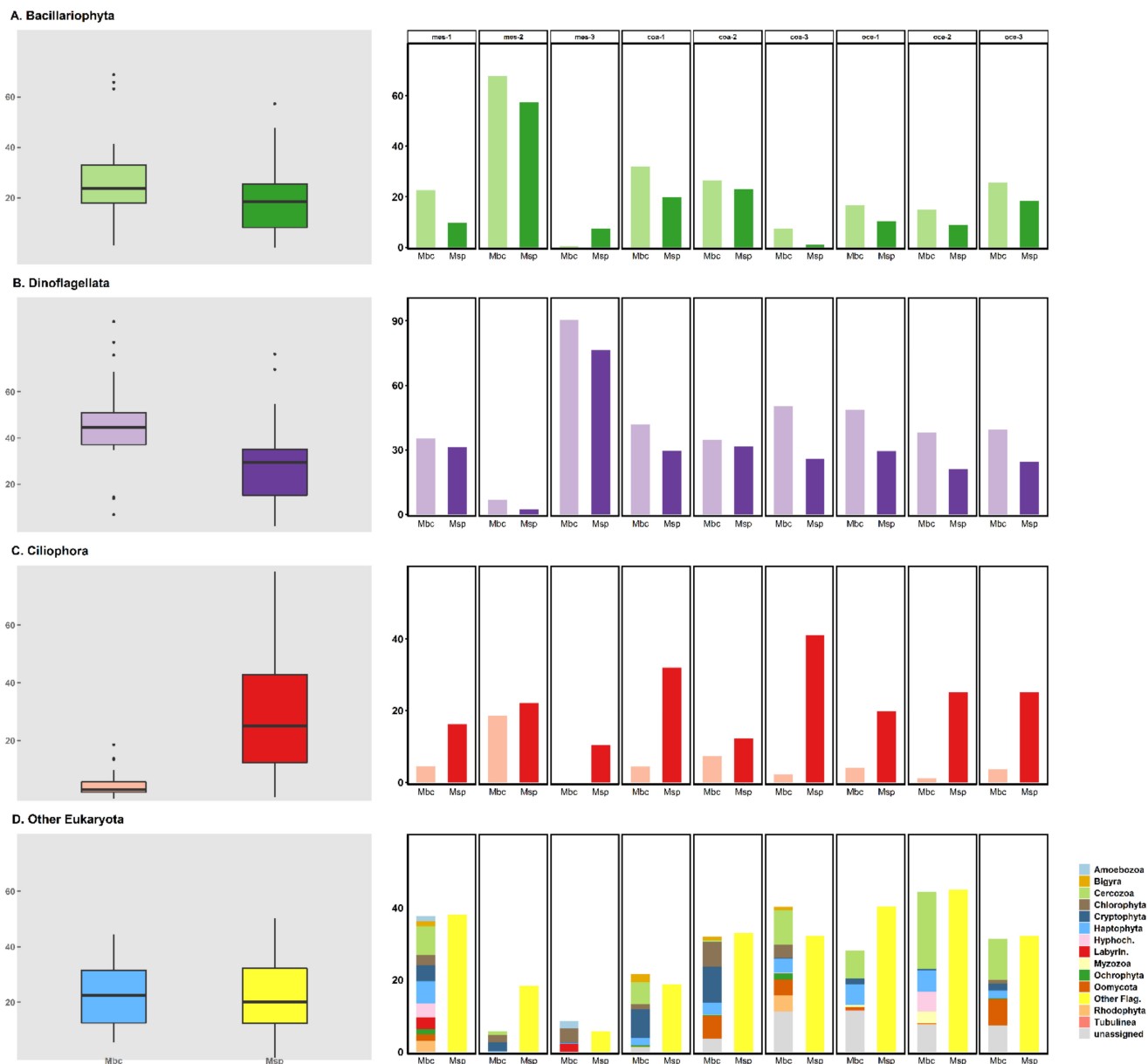

**Figure 1.** Comparison of the reads abundance percentages from DNA metabarcoding (Mbc) and the biomass percentages from microscopy (Msp). Left panel: box plots summarizing the percentages of metabarcoding reads abundance (Mbc) and biomass (Msp) out of 27 marine water samples per plankton group (**A**. Bacillariophyta, **B**. Dinoflagellata, **C**. Ciliophora, and **D**. Other Eukaryota). The vertical line inside the box plots represents the median, the top and the bottom hinges correspond to the interquartile range, and the whiskers show the minimum and maximum non-outlier values. Right panel: Analyses of Mbc and Msp percentages in nine seawater samples. Samples mes-1, mes-2, and mes-3 were from a mesocosm experiment; coa-1, coa-2, and coa-3 from coastal seawater; and oce-1, oce-2, and oce-3 from an oceanic area. The category Other Flag. includes all cells that resembled a flagellated cell but could not be identified further with optical observation. The unassigned category includes unassigned Alveolata and unassigned Stramenopiles reads.

The percentages of Ciliophora were significantly different between microscopy and metabarcoding (Figure 1C and Table 3). In this case, the relative abundance of metabarcoding reads attributed to Ciliophora was significantly lower than percentages obtained through microscopy. In Stoeck et al.'s (2014) study, the comparison of the two methodologies (microscopy and metabarcoding) concerning freshwater lake samples indicated important and statistically significant differences among morphological and taxonomical groups. Few studies have approached the diversity of plankton ciliates using either microscopical observation or metabarcoding due to the difficulties with the morphological

classification of groups [41,42] and the scientific community's persistence in analyzing phytoplankton alone. Therefore, the genetic databases contain significantly fewer entries for Ciliophora than for Dinoflagellata, as was confirmed by a manual phylum-level search in NCBI and European Nucleotide Archive (ENA) databases. Additionally, in an examination of the whole eukaryotic microbial community, recent 18S metabarcoding studies that include Ciliophora report curiously low percentages of this group [11,25,43–46]. This is irrespective of the primers, database, or bioinformatic pipeline used in each study. It is thus probable that the current 18S metabarcoding methodology is insufficient for capturing the marine Ciliophora community, and further development of primers, databases, or even pipelines may be necessary.

The percentage of groups that were microscopically characterized as flagellated cells (Other Flag.; Figure 1D) in many cases corresponded to the total percentage of reads that were characterized as Amoebozoa, Bigyra, Cercozoa, Chlorophyta, Cryptophyta, Haptophyta, Hyphochytriomycota, Labyrinthulomycetes, Myzozoa, Ochrophyta, Oomycota, Rhodophyta, Tubulinea, and unassigned groups. All the aforementioned groups include flagellated, ciliated, or amoeboid cells that are usually <20 μm in size [47]. Most modern studies classify these groups as flagellates [25,48]; this is a non-taxonomic terminology but rather an umbrella-term that groups many microbial eukaryotic taxa (Table 1). Nonetheless, even this enumeration and classification of flagellates must be accomplished by trained users who have spent a considerable amount of time observing samples [4,7]. These cell morphologies are extremely difficult to recognize optically, especially for small-size cells [42,49]. Furthermore, in many cases, cells belonging to these groups (especially Bigyra, Cercozoa, Amoebozoa, Myzozoa, and Tubulinea) are damaged during common fixation, preservation, and storage procedures, so only parts of the cells or disrupted cells are eventually observed under the microscope [16,25].This method of morphological categorization underestimates the real diversity of eukaryotic microorganisms by enumerating many different taxonomic groups (such as members of the abovementioned phyla) as members of a single non-taxonomic group (flagellated cells). Consequently, the ability to draw ecological conclusions is reduced since morphological groups cannot be related to particular ecosystem roles (for instance, heterotrophy, parasitism, and symbiosis) [47,50].

As for lower taxonomic levels of microbial eukaryotes (order and genus level; Figure 2, Dataset S2), it was observed that both techniques (morphological classification and DNA metabarcoding) detected a substantial number of common orders (exceptions to this being the results for Ciliophora). That is, a high percentage (>40%) of eukaryotic orders of Bacillariophyta and Dinoflagellata identified by microscopy were also found by metabarcoding. However, only a few Bacillariophyta and Dinoflagellata genera were recognized by both methods; for Ciliophora, these methods did not detect any genera in common because the lower the taxonomic level, the more inaccuracies resulted from morphological characterization [7,22]. For the Bacillariophyta in particular, there are cryptic species that cannot be identified optically [24,25,42]. In addition, the view of many Bacillariophyta species under the microscope may differ according to the orientation of each cell during sample preparation, usually being a completely random event [7]. The identification of Ciliophora and Dinoflagellata to the genus and species level is, by default, a demanding and troublesome task that can be accomplished only by experienced taxonomists. Another reason for not finding common taxa, especially at lower taxonomic levels, are incomplete reference databases; thus, in several cases, correct taxonomic assignment through DNA metabarcoding cannot be achieved and many taxa remain unassigned or are erroneously assigned.

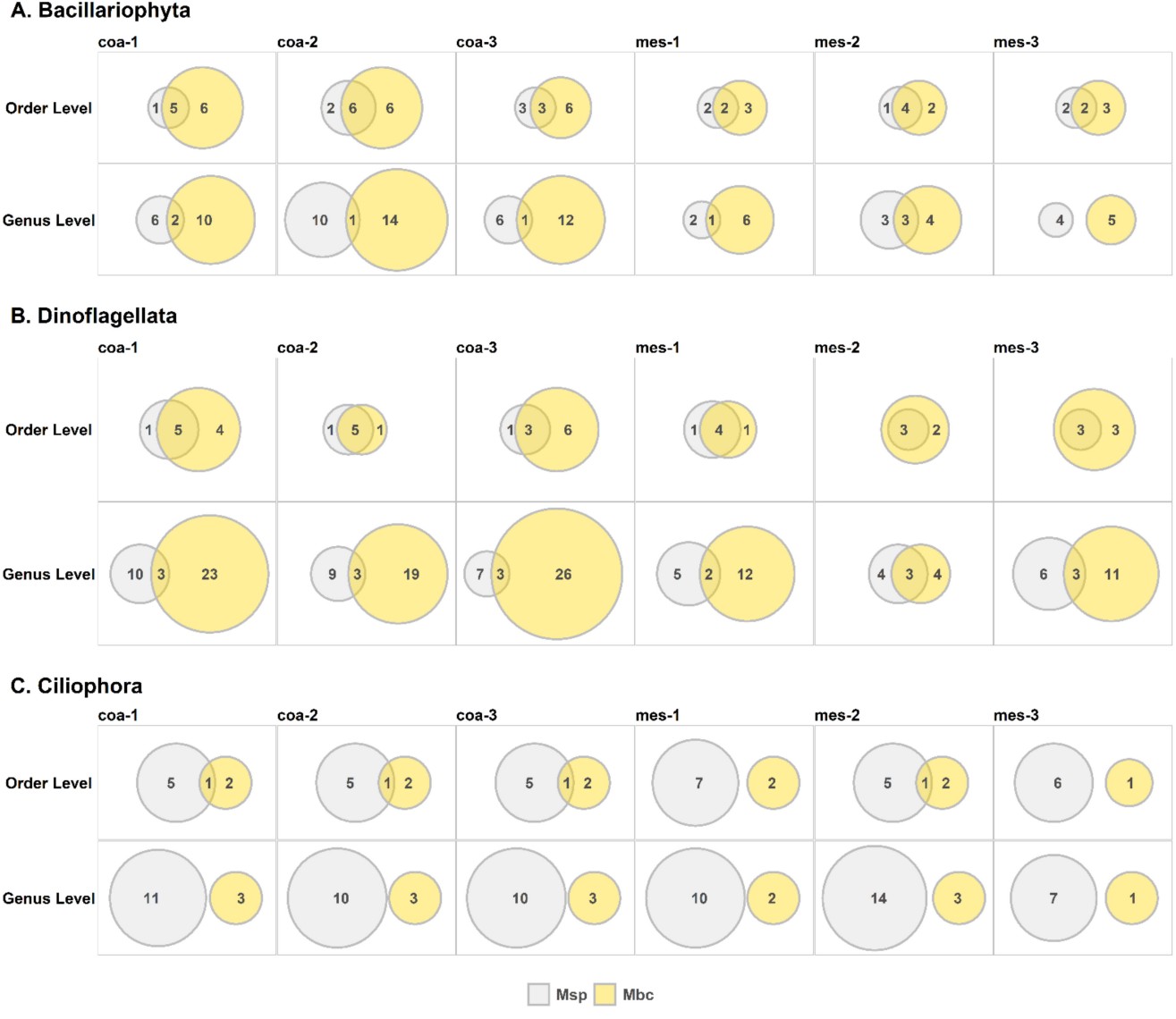

**Figure 2.** Venn diagrams indicating the number of orders or genera that resulted from microscopy (Msp in grey) and DNA metabarcoding (Mbc in yellow). The intersect area indicates the number of common orders or genera obtained through microscopy and DNA metabarcoding. The size of the circles is proportional to the number of genera or orders. Samples coa-1, coa-2, and coa-3 were from coastal seawater and samples mes-1, mes-2, and mes-3 from a mesocosm experiment.

The dependence of DNA metabarcoding on amplification through polymerase chain reaction (PCR) may introduce many technical biases that either underestimate or overestimate particular groups [51]. In addition, to give DNA metabarcoding results an ecological meaning, it is important that curated high-quality reference databases with high entry numbers of the targeted sequences are used for the taxonomic assignment [51]. Another critical point is that the number of copies of the targeted sequences contained in a microorganism remains unknown [42], which could also create biases in the estimated relative frequencies of the different taxa in a sample. DNA metabarcoding may not yet produce precise quantitative results, and comparisons between different metabarcoding studies encompass many pitfalls, especially due to the different methodologies and taxonomic frameworks that have been developed for photoautotrophic and heterotrophic eukaryote groups [42]. Metabarcoding data can be more comparable when a standardized protocol that includes positive controls, such as mock communities, is used, even though biases may still occur between different laboratory pipelines and sequencing runs.

Morphological characterization has a much lower cost for consumables compared with that of DNA metabarcoding. However, optical observation demands more laboratory time until the final dataset is obtained. In this study, 50 working days were required to acquire the final datasets from the optical analyses for 100 samples, without calculating the sampling procedure, which is common for the two methodologies. In contrast, 19 working days were required to obtain the final datasets for 100 samples using DNA metabarcoding. Thus, the personnel cost for the microscopical identification is considerably higher than that for DNA metabarcoding, rendering the total cost of the two methodologies comparable.

In terms of the well-studied and easily recognizable groups of microorganisms (for example, Dinoflagellata and Bacillariophyta), morphological classification may produce reliable quantitative results at least to the order level, whereas the experience of the user increases the credibility for the lower taxonomic levels. However, experienced researchers able to recognize low taxonomical levels (genus, species) are scarce and becoming even scarcer with time; thus, the microscopical identification of low taxonomic levels becomes a risky task as results from inexperienced researchers are not trustworthy.

Through DNA metabarcoding, more taxonomic groups are recognized with higher accuracy [22,23,25,42,43,52], and the present study highlights that metabarcoding is a more accurate approach for the assessment of the biological richness of an ecosystem. However, metabarcoding may under- or overestimate the abundance of microorganisms, so the quantitative results of morphological observation may be more reliable for many groups. It seems that through DNA metabarcoding, we may obtain accurate results when asking the ecological questions about who is present in an ecosystem, whereas microscopy may better answer questions regarding the abundance in an ecosystem.

## 4. Conclusions

This study showed that microscopy provides precise quantitative results but only for certain groups, and it has low taxonomic resolution, usually at the order level. DNA metabarcoding can only provide semi-quantitative results, usually with biases; however, it offers much higher and more accurate taxonomic resolution, which can be further improved both by better PCR primer selection and by complementing the reference databases.

Since the two methods can be complementary, their combined use in the study of marine microbial eukaryotes would provide much more reliable and accurate results. The challenge is to standardize an analytical pipeline where samples can be processed smoothly by both methodologies and their results combined to more accurately present eukaryotic microbial diversity, both qualitatively and quantitatively. When these techniques are to be used separately, the advantages and disadvantages of each should be considered and described. The decision of whether to use one technique or the other, or a combination of both should be made based on the questions each study aims to answer in relation to the advantages and disadvantages of each method.

**Supplementary Materials:** The following are available online at https://www.mdpi.com/article/10.3390/d13050180/s1. Dataset S1: Dataset containing the reads percentages from DNA metabarcoding (Mbc) and the biomass percentages from microscopy (Msp), Dataset S2: Dataset containing the reads percentages from DNA metabarcoding (Mbc) for the category "Other Eukaryota".

**Author Contributions:** Conceptualization, I.S. and P.K.; formal analysis, I.S.; funding acquisition, I.K. and P.P.; methodology, I.S., P.K. and P.P.; writing—original draft, I.S.; writing—review and editing, I.S., P.K., I.K. and P.P. All authors have read and agreed to the published version of the manuscript.

**Funding:** The sample collection of this work was funded in the frame of the Study of the marine and submarine environment and ecosystem of the East Mediterranean, south of Crete Island project financed by the General Secretariat for Research and Technology (GSRT) through the Programming Agreements between Research Centers—GSRT2015—2017/22.4.2015, and by the European Union and Greek national funds through the Operational Program Education and Lifelong Learning of the National Strategic Reference Framework (NSRF)—ARISTEIA II (HYPOXIA project, No 5381). Additionally, this work was supported by the project Blue Growth with Innovation and application in

the Greek Seas—GLAFKI (MIS 5002438) funded by national and EU funds under National Strategic Reference Framework 2014–2020 and by the project Centre for the study and sustainable exploitation of Marine Biological Resources (CMBR) (MIS 5002670), which is implemented under the action Reinforcement of the Research and Innovation Infrastructure, funded by the Operational Program Competitiveness, Entrepreneurship and Innovation (NSRF 2014–2020) and co-financed by Greece and the European Union (European Regional Development Fund). This research was supported in part through computational resources provided by IMBBC (Institute of Marine Biology, Biotechnology and Aquaculture) of the HCMR (Hellenic Centre for Marine Research). Funding for establishing the IMBBC HPC has been received by the MARBIGEN (EU Regpot) project, LifeWatchGreece RI and the CMBR (Centre for the study and sustainable exploitation of Marine Biological Resources) RI.

**Institutional Review Board Statement:** Not applicable.

**Informed Consent Statement:** Not applicable.

**Data Availability Statement:** The data presented in this study are available in the Supplementary Materials.

**Acknowledgments:** We are grateful to Georgia Assimakopoulou for providing microscopy data from the oceanic site and to Manolis Ladoukakis for critical discussions.

**Conflicts of Interest:** The authors declare no conflict of interest.

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
