# Peer review of "A Comparison of DNA Metabarcoding and Microscopy Methodologies for the Study of Aquatic Microbial Eukaryotes"

_diversity, doi:10.3390/d13050180_

Round 1
Reviewer 1 Report
Santi and colleagues aimed to compare the microscopy-based identification of aquatic microbial eukaryotes with results from DNA metabarcoding. I appreciate their efforts to indicate critical points of applying methods and I believe that the Authors did a good job to examine the advantages and disadvantages of each methodology. Moreover, this paper is really interesting; however, the Authors applied previously published data (Santi et al. 2019, Santi et al. 2020) and conclusions from presented work are similar to Piredda et al. 2017 so that it seems that here is almost nothing new.
For future experiments, I would suggest to perform additional analysis, i.e., i) check how the choice of the primer set directly affects the relative abundances of aquatic microbial eukaryotes DNA metabarcoding; ii) perform DNA metabarcoding of sample consisted with the species which were identified on the morphological level - mock control samples should be applied to know what is truly aquatic microbial eukaryotes diversity in the samples. In this way you could also check sequences deposited in the GenBank database: i) are they correctly label to the target species?, ii) did you received overestimated species diversity in your HTS data?
Unfortunately, I am recommending rejection at this time until extensive revisions with a new analysis can be made. I hope that the Authors see my comments as an effort to help them improve presented study.
Piredda et al. 2017, doi:10.1093/femsec/fiw200.
Santi et al. 2019, doi:10.1016/j.marenvres.2019.104752. 365
Santi et al. 2020, doi:10.3354/ame01933.
Reviewer 2 Report
This study describes a comparison between the morphological (microscopy) and molecular (metabarcoding) assessment of microbial eukaryotic communities. The methods used are appropriate for the objectives described, and the manuscript is written in a straightforward and understandable manner. The figures illustrate the results in a clear way. The study provides an important contribution as few other have compared the aforementioned methodologies for the assessment of biodiversity in these taxa.
My main comment and concern is a significant lapse in the description of the Materials & Methods. These should be descriptive enough to be able to reproduce the results, which is far from true in the current state of the MS. The samples are inadequately described, as is the molecular and bioinformatic pipeline. The authors call for the need for standardised protocols while they have failed to provide their own in the MS. Thus, the M&M section requires substantial additions. Some conclusions are not fully supported by the results, unless I have made a mistake in my interpretation, which could of course be the case. Table and Figure captions require minor revisions. Specific line-by-line comments are listed below.
L38: Correct to “planktonic eukaryotes”.
L40: Correct to “necessity for”.
L44: “fixed” as in fixated using a fixative or as in a sample of “fixed” (=constant) size? I would rephrase for clarity.
Table1: Microscopy groups: Capitalise first letter.
L55: Do the prefixes zoo- and phyto- relate to physiology of these groups? Or rather their function?
L60: Why are these groups omitted?
L66: Add comma before “others”.
L67: Is there a difference between microplankton and plankton? If so, please elaborate; if not, adhere to one of the 2 terms consistently.
L73-74: It is my understanding that “microbial eukaryotes” and “microbial plankton” are being used interchangeably; this is confusing to the reader; adhere to either term consistently.
L75: Correct to “to provide”.
L79: “Open sea” and “coastal areas” is not specific enough; which sea? How many samples from each environment?? State clearly where these samples were taken as this is important information. I would replace Table 1 with one describing the sampling localities (latitude, longitude, depth, date of sampling).
L85: Correct to “a geometric shape was assigned to each cell”.
L85: What kind of filtration? What volume? What size filters? How much time spent filtrating? This is hardly enough information to reproduce the methods.
L88-89: Referencing these publications does not suffice; the M&M should provide enough information for protocols to be reproduced. There is no information on how the DNA was handled for library preparation, nor the bioinformatic analyses. This is very important information! Which primers? PCR mix? How many reads produced? What algorithm was used? ASVs or OTUs? What values were set for different parameters? How was taxonomy assigned? Are these data rarefied? The study is a comparison of morphology and metabarcoding, though the latter is not even described at all. These details must be included here.
L101-106: The data analysis description is lacking. What software was used and which packages? What was the linear model formula? Why was beta distribution selected? What assumptions were made for the data? How were these checked? All this information needs to be included.
L110: Delete “found”.
L115: “Statistically significant resemblance”? Either there are statistically significant differences, or there is an absence thereof. Statistically significant resemblance is not valid here I believe.
L120: In contrast to? Which size fraction was studied here? This is not stated in M&M.
L129: Correct to “morphological identification exclusively”.
L139: Delete “that”.
L185: Correct to “ecosystem roles”.
L187-192: I do not think that the results support the statement that a “substantial” number of identified orders were common, or “>50%”, at least not for Bacillariophyta. Unless I am mistaken: e.g. coa-2: Total orders=2+6+6=14; common=6/14=43%. The above statements could be made for Dinoflagellata only.
L199: Correct to “identification”.
L202: Correct to “are incomplete reference databases”.
L207: Explain this statement.
L217: What is meant here by “positive controls”?
L219: Correct to “Morphological characterization has a much lower cost for consumables compared to that of DNA metabarcoding”.
L222: 100 samples? While 28 were used in the model? Please explain.
L231: The word “user” is inappropriate; replace with “researcher” or “taxonomist”.
L234: Explain why this is risky? Because the taxonomic assignments are likely to be less trustworthy?
L253: Delete this first sentence and focus on the conclusions. Begin with something in the line of “Our study has shown that microscopy gives precise quantitative results but only for certain groups, and it has low taxonomic resolution usually at the order level.”
L259: Delete “respective”.
L263: Replace “swimmingly” with “smoothly”.
L264: Replace “in case” with “when”.
L268: End sentence at “method”. “
L269-272: Delete this sentence.
Table2 caption: Typo for “metabarcoding”. The number of samples used is not information for a table caption; this should be in M&M. “Other flag.” is not defined in the main text; this should be included.
Figure 1 caption: The details provided in the caption describing the samples should be included in M&M.
Figure 2 caption: Delete “i.e. the orders or genera that both methods identified”, this is obvious from the previous part of the sentence. Include the colours that describe each methodology (e.g. “Msp in gray”); include that the size of the circle is proportional to the number of genera or orders.
Reviewer 3 Report
Dear authors, please find minor comments and edits to the paper.
Thank you for your work.
